# Intrauterine Fetal Demise After Uncomplicated COVID-19: What Can We Learn from the Case?

**DOI:** 10.3390/v13122545

**Published:** 2021-12-19

**Authors:** Pavel Babal, Lucia Krivosikova, Lucia Sarvaicova, Ivan Deckov, Tomas Szemes, Tatiana Sedlackova, Michal Palkovic, Anna Kalinakova, Pavol Janega

**Affiliations:** 1Department of Pathology, Faculty of Medicine, Comenius University, 811 08 Bratislava, Slovakia; lucia.krivosikova@fmed.uniba.sk (L.K.); michal.palkovic@fmed.uniba.sk (M.P.); pavol.janega@fmed.uniba.sk (P.J.); 2Department of Pathology, University Hospital, 917 75 Trnava, Slovakia; lucia.kolenova@gmail.com; 3Department of Gynecology and Obstetrics, University Hospital, 917 75 Trnava, Slovakia; ivan.deckov@gmail.com; 4Laboratory of Genomics and Bioinformatics, Comenius University Science Park, 841 04 Bratislava, Slovakia; tomas.szemes@uniba.sk (T.S.); tatiana.sedlackova@uniba.sk (T.S.); 5Health Care Surveillance Authority, 829 24 Bratislava, Slovakia; 6Public Health Authority of the Slovak Republic, 826 45 Bratislava, Slovakia; anna.kalinakova@uvzsr.sk; 7Centre of Experimental Medicine, Institute of Normal and Pathological Physiology, Slovak Academy of Sciences, 813 71 Bratislava, Slovakia

**Keywords:** SARS-CoV-2, fetal demise, syncytiotrophoblast, persistent infection, transplacental transmission

## Abstract

Background: SARS-CoV-2 infection in pregnant women can lead to placental damage and transplacental infection transfer, and intrauterine fetal demise is an unpredictable event. Case study: A 32-year-old patient in her 38th week of pregnancy reported loss of fetal movements. She overcame mild COVID-19 with positive PCR test 22 days before. A histology of the placenta showed deposition of intervillous fibrinoid, lympho-histiocytic infiltration, scant neutrophils, clumping of villi, and extant infarctions. Immunohistochemistry identified focal SARS-CoV-2 nucleocapsid and spike protein in the syncytiotrophoblast and isolated in situ hybridization of the virus’ RNA. Low ACE2 and TMPRSS2 contrasted with strong basigin/CD147 and PDL-1 positivity in the trophoblast. An autopsy of the fetus showed no morphological abnormalities except for lung interstitial infiltrate, with prevalent CD8-positive T-lymphocytes and B-lymphocytes. Immunohistochemistry and in situ hybridization proved the presence of countless dispersed SARS-CoV-2-infected epithelial and endothelial cells in the lung tissue. The potential virus-receptor protein ACE2, TMPRSS2, and CD147 expression was too low to be detected. Conclusion: Over three weeks’ persistence of trophoblast viral infection lead to extensive intervillous fibrinoid depositions and placental infarctions. High CD147 expression might serve as the dominant receptor for the virus, and PDL-1 could limit maternal immunity in placental tissue virus clearance. The presented case indicates that the SARS-CoV-2 infection-induced changes in the placenta lead to ischemia and consecutive demise of the fetus. The infection of the fetus was without significant impact on its death. This rare complication of pregnancy can appear independently to the severity of COVID-19’s clinical course in the pregnant mother.

## 1. Introduction

The two opening decades of the third millennium brought to humans three serious infections caused by coronaviruses, the last becoming a world pandemic spreading to every corner of the planet. The disease, named COVID-19 and caused by the virus SARS-CoV-2, predominantly afflicts the respiratory system, with effects on other tissues and organs, including the placenta. Pregnant women suffer from the same symptoms as non-pregnant women, such as fever, cough, and myalgia, but are more susceptible to the poor outcomes of COVID-19 [1,2] and pregnancy complications leading to stillbirth, premature rupture of membrane, or intrauterine fetal demise (IUFD) [3,4]. The possibility of vertical transmission of SARS-CoV-2 infection during pregnancy has been considered and remains an issue under discussion [5,6,7,8], with a number of studies that did not confirm this process [9,10,11].

One of the most serious complications of pregnancy in mothers with COVID-19 is intrauterine fetal death. The reasons for this occurrence are not unambiguous, but pathological changes to the placenta, especially blood perfusion reduction in placental tissue [4,10,12,13] and viral infection of the fetus might be considered [5,14]. IUFD remains an infrequent complication of pregnancy with COVID-19, without an overall dramatic increase in incidence of this event compared with the pre-COVID period [15]. Here, we present a case of IUFD that happened in the 38th gestational week, providing a detailed analysis of possible factors participating in this unwished complication of pregnancy.

## 2. Materials and Methods

### 2.1. Histology

Tissues were fixed in 10% formalin and submitted to standard routine paraffin processing. Histological slices were stained with hematoxylin and eosin for basic and diagnostic histological evaluation. Staining with naphthol-AS-D-chloroacetate for the detection of chloroacetate esterase (CHAE) activity was performed to evaluate tissue infiltration by granulocytes. A phosphotungstic acid hematoxylin (PTAH) stain was used to evaluate the presence of fibrin.

### 2.2. Immunohistochemistry

The formalin-fixed, paraffin-embedded tissues (FFPE) were cut to 5 um thick slices, which were deparaffinized, rehydrated, treated for antigen retrieval and immunohistochemically stained using antibodies against the following proteins: CD3, CD4, CD8, CD20, CD68 (prediluted antibodies, Agilent Technologies, Santa Clara, CA, USA), SARS-CoV-2 nucleocapsid protein, Angiotensin-converting enzyme 2, ACE2, Transmembrane Serine Protease 2 (TMPRSS2), Basigin/CD147 (prediluted antibodies, Bio-SB, Santa Barbara, CA, USA), PDL-1 (dilution 1:500, ab205196; Abcam, Cambridge, UK), and SARS-CoV-2 spike S1 protein (dilution 1:50; Sino Biological, Inc., Beijing, China). After 1 h incubation of the primary antibody at room temperature, the immunohistochemical reaction was visualized by immuno-peroxidase technique using the FLEX EnVision Kit (Agilent Technologies, Santa Clara, CA, USA) in a DAKO Autostainer (Dako, Glostrup, Denmark). For double staining, the Histofine kit for immuno-alkaline phosphatase technique was applied (Nichrei Biosciences Inc., Tokyo, Japan).

### 2.3. In Situ Hybridization

For detection of the SARS-CoV-2 virus’ RNA in histological sections, the RNA Scope^®^ detection system, with the use of probes for genomic RNA coding of nucleocapsid (846081) and spike (848561) proteins, was used (Advanced Cell Diagnostics, Inc., Newark, CA, USA).

### 2.4. qRT PCR SARS-CoV-2 Detection

To verify the presence of the SARS-CoV-2 virus in the placenta sample, nucleic acids were isolated from the FFPE sample of placenta tissue using The ReliaPrep™ FFPE gDNA Miniprep System (Promega Corporation, Madison, WI, USA) according to the original protocol, omitting the RNase A digestion step to extract potential virus RNA. After that, qPCR analysis was performed, with the nucleic acids’ sample as a template. For the preparation of the qPCR reaction, the qb SARS-CoV-2 Multiplex EndoC kit (GENERI BIOTECH, Hradec Kralove, Czech Republic) was used, followed by detection on the BioRad CFX Connect Real-Time PCR system (BioRad Laboratories, Dubai, United Arab Emirates). The qPCR reaction was performed in duplicate.

Detection of SARS-CoV-2 in fresh tissue samples was performed using the following procedure: the RNA extraction was carried out using RNeasy Plus Mini Kit (Qiagen GmbH, Hilden, Germany) according to protocol for tissue samples and with the use of TissueRuptor II (Qiagen GmbH, Hilden, Germany) for tissue disruption and homogenization. The same SARS-CoV-2 RT PCR assay was used as described in the previous paragraph.

## 3. Case Study

### 3.1. Clinical Course

A 32-year-old Caucasian woman at the 38th week of gestation was admitted to the gynecology department on 21 February 2021, at 11:49 p.m., stating an absence of fetal movements since 7:00 p.m. and the presence of contractions since 8:00 p.m., and every 5 min at the time of admission. Sonography identified asystole of the heart of the fetus without pulsations in the umbilical cord, and placental grading was II with a normal value of amniotic fluid. Soon after the admission, at 11:55 p.m., she spontaneously delivered a boy of 3315 g/51 cm, Apgar score 0. The placenta separated completely and was expulsed according to Baudelocque-Schultze.

The gynecological history of the patient was without notability. This was her third pregnancy; her last menstrual cycle was on 29 May 2020. Ultrasonographies on the 13th, 20th, and 31st gestational week documented appropriate fetal growth and did not uncover developmental defects.

She overcame COVID-19 with mild symptoms, without the necessity of hospitalization. Her PCR test from a nasal swab on 31 January 2021 was positive.

Laboratory results: Blood WBC 10^9/l: 13.1, S-CRP mg/L: 11.51; Global coagulation tests: P_PT%: 120.6; P_PT R *: 0.93; P_INR *: 0.91; P_aPTT s: 23.0; P_aPTT R *: 0.96 were normal.

### 3.2. Pathological Examination of the Placenta

#### 3.2.1. Gross Findings

The placenta had a size of 16 × 16 × 3 cm and weight of 480 g, corresponding to the gestational age. The three-vessel umbilical cord was inserted centrally, was of a reddish color, 48 cm long, with 1–2 right twists/10 cm and mild varicosities, without real nods. The membranes were slightly yellowish-green, with light peeling-off. The maternal surface was complete, with multiple geographic, merging, grayish-white areas with small calcifications, affecting up to 70% of the placental tissue. These lesions were macroscopically visible on the cut section, with higher frequency underneath the fetal surface (Figure 1).

#### 3.2.2. Histology

The umbilical cord contained three vessels embedded in Warthon´s jelly, was without inflammatory infiltrate, and the lumina were without thrombosis. The membranes were of normal histological structure and the amniotic epithelium showed reactive vacuolization of cytoplasm. Decidua capsularis was lightly infiltrated by a mixed inflammatory cell population.

The histological picture of the placenta corresponded to the terminal stage of gestation (38+ GW).

Pathological changes represented the massive diffuse distribution of foci with accumulation of intervillous fibrin, villous clumping, and villous necrosis in central parts of the foci, with small calcifications. The chorionic villi at the periphery of the foci showed interstitial edema and infiltration with scant histiocytes and lymphocytes, neutrophils appeared only in the necrosis in infarctions, the blood vessels were dilated with congestion, and the syncytiotrophoblast showed sporadic nodule formation. These changes were accompanied by massive fibrin deposition, detected by phosphotungstic acid hematoxylin in the intervillous space, with various degrees of lymphohistiocytic infiltration with an admixture of scant neutrophils (Figure 2).

#### 3.2.3. Immunohistochemistry

Immunohistochemical detection of SARS-CoV-2 nucleocapsid protein and spike protein showed only foci with strong cytoplasmic expression in syncytiotrophoblast cells. These positive foci were in centers of the affected areas with necrosis and in isolated foci of morphologically unaffected villi (Figure 2 and Figure 3). The expression of ACE2 was weak, and focal in decidual cells and the syncytiotrophoblast. TMPRSS2 protein expression was weak, barely detectable, in fact, and exactly the same as the expression in control placental tissues. There was diffuse moderate to strong positivity of CD147 protein and strong expression of PDL-1 protein on the syncytiotrophoblast cell membrane (Figure 3).

Histiocyte-macrophages (CD68+) and T-lymphocytes (CD3+; CD4:CD8/1:4) were present in the individual inflammatory cell infiltrates in the villus interstitium, but were numerous in the intervillous fibrin deposits, with prevalence of histiocyte-macrophages together with a few neutrophils. Double staining of SARS-CoV-2 and inflammatory cells did not show increased accumulation of lymphocytes or macrophages in areas with SARS-CoV-2 protein positivity (Figure 2).

#### 3.2.4. In Situ Hybridization Detection of SARS-CoV-2 RNA

In-situ hybridization, with the probes for genomic RNA coding nucleocapsid and spike proteins, identified individual or small groups of positive cells in the intervillous space or in the syncytiotrophoblast (Figure 3). These positive areas did not align with the immunohistochemical distribution of nucleocapsid and spike proteins. No hybridization signal was observed in cells within placental villi.

#### 3.2.5. qRT-PCR of SARS-CoV-2 RNA

The qPCR analysis of formalin fixed tissue confirmed the presence of the SARS-CoV-2 virus. The signal for the virus was detected for both analyzed E and RdRp virus genes, with mean Ct values of 28.34 and 28.68, respectively, representing Ct values at the level of the positive control.

### 3.3. Autopsy of the Fetus

#### 3.3.1. Gross Findings

The fetus was a mature boy, grossly without anatomical defects, 51 cm long and weighing 3150 g. The autopsy was performed 32 h after delivery. All organs and tissues were of appropriate position, shape, and structure.

#### 3.3.2. Histology, Immunohistochemistry, and In Situ Hybridization

Histological examination of slides stained with hematoxylin and eosin showed normal histomorphological structure. A mild increase in interstitial infiltration by small lymphocytes was noted in the lungs (Figure 4).

Immunohistochemical examination of the lung tissue uncovered diffuse interstitial infiltration by equal populations of CD20+ B-lymphocytes and CD3+ T-lymphocytes, with a prevalence of CD8+ over CD4+ cells with a 4:1 ratio. The density of both T- and B-lymphocyte infiltrate was more than four-fold higher compared with control lungs of newborns (Figure 5). No remarkable increase in CD68+ macrophages and neutrophils was detected. Scattered individual or small clusters of SARS-CoV-2 nucleocapsid protein-positive pulmonary alveolar epithelial cells and individual blood vessel endothelial cells were detected in the lungs (Figure 4). Few individual endothelial cells of the pulmonary vessels were also positive for the presence of SARS-CoV-2 by in situ hybridization with the probe for nucleocapsid protein-coding RNA (Figure 4). The potential virus-receptor proteins ACE2, TMPRSS2, and CD147 were not detected in the lung tissue, except for CD147 expression in the bronchial and bronchiolar epithelial cells (data not shown). The lung tissue of the control newborn showed similar TMPRSS2 and CD147 expression pattern; however, ACE2 was positive in vascular endothelial cells. PDL-1 was expressed in alveolar macrophages, with significantly weaker positivity in the fetal lungs (data not shown).

Immunohistochemical analysis of other tissues and organs, except for the kidney, did not show the presence of SARS-CoV-2 nucleocapsid or spike proteins. SARS-CoV-2 nucleocapsid protein was detected in a few individual epithelial cells of the proximal convoluted tubule; the potential virus-receptor proteins ACE2, TMPRSS2, and CD147 were diffusely positive in both the proximal and the distal convoluted canaliculi epithelial cells (data not shown).

#### 3.3.3. qRT PCR

Fresh tissue samples from the lung, nasopharynx, and umbilical cord were positive for the presence of SARS-CoV-2 RNA, detected by qRT-PCR for both analyzed E and RdRp virus genes, with mean Ct values of 37.57, 32.54, and 32.17, respectively.

## 4. Discussion

The presented case of fetal demise in a pregnant woman with confirmed COVID-19, but without other noteworthy clinical or obstetric disorders, indicates that fetal death is a possible consequence of SARS-CoV-2 infection in pregnancy. SARS-CoV-2 infected pregnant women are at higher risk of maternal and fetal adverse outcome; however, the fetal outcome does not seem to depend on the clinical form of COVID-19 symptoms of the mother [16]. An interplay of several factors may participate in such an unfavorable conclusion of pregnancy. As also stated by others [7,14,17,18], pathological changes in both the placenta and the fetus may result in fetal demise. Some findings in the presented case open additional questions about the pathogenesis of the affection of the placenta and the fetus.

The dominant histopathological finding in the placenta was the massive intervillous deposition of fibrinoid, leading to clumping of neighboring villi up to the formation of large aggregates with central necrosis, corresponding to placental infarction. Histological changes identified in our case were consistent with the morphological criteria for standardized definition of placental infection by SARS-CoV-2, suggested by Roberts et al. [19]. Similar findings of an extensive affection of the placenta have been reported in some patients [6,14,19,20], but not in others [9,13], and there was no correlation with the severity of the clinical course of the COVID-19 disease in the pregnant patient. The fibrinoid deposits may be accompanied by an inflammatory reaction with predominant macrophage infiltration [2,13], as was also seen in our case. The intense accumulation of polymorphonuclear leukocytes was detected in some cases [20,21], indicating that activation of NETosis could be part of the developing pathology [6]. In the majority of reported cases, including the one presented here, no prominent villitis was observed.

The infection of tissues by SARS-CoV-2 can be identified by immunohistochemical detection of the virus-related proteins [4,6,22]. Multiple works documented diffuse staining of syncytiotrophoblast cells in the placenta from some COVID-19-diagnosed patients [6,23], however, in some case series, only a few or no positive cases were detected [6,13]. Surprisingly, in our case, only isolated groups of syncytiotrophoblast cells showed immunohistochemical positivity for SARS-CoV-2 nucleocapsid and/or spike proteins. It is noteworthy that the virus’ footprints were detectable over three weeks after the positive PCR test in the patient with a mild form of COVID-19. The time factor may play a role in the extent of placental positivity, since the diffuse positivity of the placenta was detected in specimens examined less than 14 days after the mother´s COVID-19 positive PCR test from the nasopharyngeal swab [6,11,23]. The viral presence in the placenta in our case was also confirmed by PCR test and in situ hybridization. Although in situ hybridization detected the virus only in scattered individual cells, the use of an anti-sense probe documented the presence of a replicating virus [19]. The in situ hybridization results did not overlap with the immunohistochemical detection of the viral proteins. We may speculate that the elimination of the proteins is delayed after the virus’ cessation.

In immunocompetent individuals, the SARS-CoV-2 replication with the shedding of infectious viral particles does not exceed two weeks [24]. The placenta represents an embryonic tissue foreign to the host organism, that secures immunotolerance from maternal immunocompetent cells by expressing the immunity check-point PDL-1 protein, leading to the inhibition of helper T-lymphocytes. Ineffectiveness of the PD-1/PDL-1 axis might be involved in persistent viral infection [25]. This could explain the permanent synthesis of the viral proteins that continue to induce the coagulation system [26], resulting in a large extent of intervillous fibrinoid depositions and placental infarction formation, as was seen in the presented case. Continuous activation of complement parallel to the intervillous inflammation may represent another factor promoting the placental tissue damage [21].

It is generally accepted that SARS-CoV-2 spike protein binds to its receptor, human ACE2 (hACE2), through its receptor-binding domain (RBD) and is proteolytically activated by human proteases. These SARS-CoV-2 entry-activating proteases include cell surface protease TMPRSS2 and lysosomal proteases cathepsins [27]. Another molecule, basigin2 (CD147), was suggested to be involved in SARS-CoV-2 tropism and has been shown to mediate the virus’ entrance into host cells by endocytosis [28,29]. The detection of low levels of ACE2 and TMPRSS2 in the placenta was in contrast with the high expression of CD147 on the syncytiotrophoblast, indicating that this protein could serve as the dominant receptor for the virus, participating at the initiation and persistence of placental tissue SARS-CoV-2 infection.

The transplacental transfer of SARS-CoV-2 has been discussed by numerous authors in recent months. Several studies did not confirm any transplacental transfer, even in a large patient series [11,13,30], while others demonstrated placental infection with the virus’ transfer to the fetus [6,14,18,23]. There is sufficient evidence, also supported by our case, that transplacental SARS-CoV2 transfer to the fetus may happen, although this complication, fortunately, appears to be rare. Post mortem immunohistochemical detection of the SARS-CoV-2 proteins is limited by the time period between death and autopsy. The autopsy of the fetus, performed 34 h after death, was at the upper limit for reliable immunohistochemical SARS-CoV-2 detection possibility [22]. However, the infection of the fetus was also confirmed by the in situ hybridization and the qRT-PCR. In spite of the SARS-CoV-2 infection of the fetus, no histomorphological signs of pulmonary tissue damage were observed. In the lungs, mild to moderate interstitial lymphocytic infiltrate was the only pathological finding.

## 5. Conclusions

Findings in the presented case indicate that fetal SARS-CoV-2 virus infection was not the cause of the fetal demise. The persistence of the virus proteins’ expression by the trophoblast lead to extensive intervillous fibrinoid deposition, with consecutive placental infarction formation and ischemia that lead to the fetus’ demise. This rare complication of pregnancy can appear regardless of the severity of COVID-19’s clinical course in the pregnant woman.

## Figures and Tables

**Figure 1 viruses-13-02545-f001:**
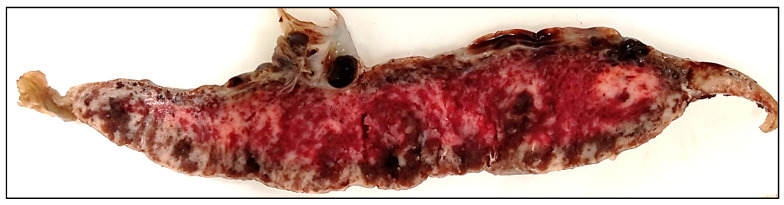
Cut surface of the placenta showed numerous merging grayish-white areas of infarctions and dark red foci with hemorrhage.

**Figure 2 viruses-13-02545-f002:**
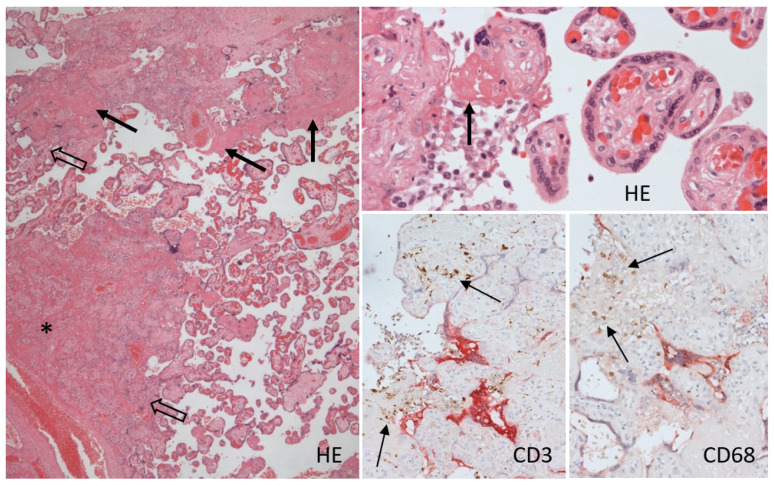
Histological evaluation of the placenta showed large-scale depositions of intervillous fibrinoid (arrow) with clumping of chorionic villi (empty arrow) and necrosis in the centers of large agglomerates (*). Double staining immunohistochemistry showed focal groups of syncytiotrophoblast cells expressing SARS-CoV-2 nucleocapsid protein (red staining) accompanied by increased accumulation of T-lymphocytes (CD3) and macrophages/histiocytes (CD68) (brown staining), especially in the intervillous fibrinoid (less in the villi). Hematoxylin and eosin, 25×, 200×; SARS-CoV-2 Nc protein: immuno-alkaline phosphatase (red), CD3, CD68: immunoperoxidase, diaminobenzidine (brown), 100×.

**Figure 3 viruses-13-02545-f003:**
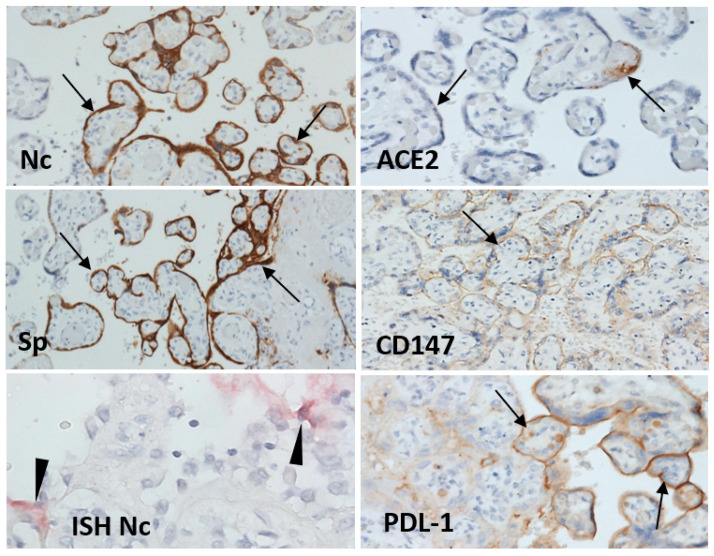
Placenta showed focal expression of SARS-CoV-2 nucleocapsid (Nc) and spike (Sp) protein in syncytiotrophoblast (arrow) and SARS-CoV-2 viral RNA in scattered individual trophoblast cells (arrowhead). On syncytiotrophoblast, there was weak to moderate focal ACE2 and diffuse expression of basigin (CD147), a tentative binding receptor for SARS-CoV-2 spike protein; PDL-1 expression was diffuse and strong. Immunoperoxidase, diaminobenzidine, 200x; in-situ hybridization (ISH), 600x.

**Figure 4 viruses-13-02545-f004:**
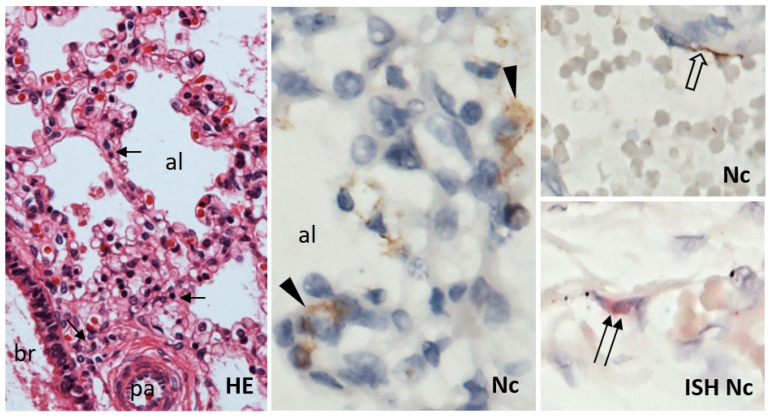
Lungs of the fetus had normal histological structure of bronchi (br), pulmonary arteries (pa), and alveoli (al); only a slight increase in small lymphocyte infiltrate (arrow) in the interstitium was noted. Small groups or individual alveolar epithelial cells (arrowhead) and a few individual endothelial cells (open arrow) showed immunohistochemical positivity of SARS-CoV-2 nucleocapsid protein (Nc). Scattered endothelial cells were positive for the presence of SARS-CoV-2 RNA by in situ hybridization (ISH; double arrow). HE, 200×; Nc, ISH Nc, 600×.

**Figure 5 viruses-13-02545-f005:**
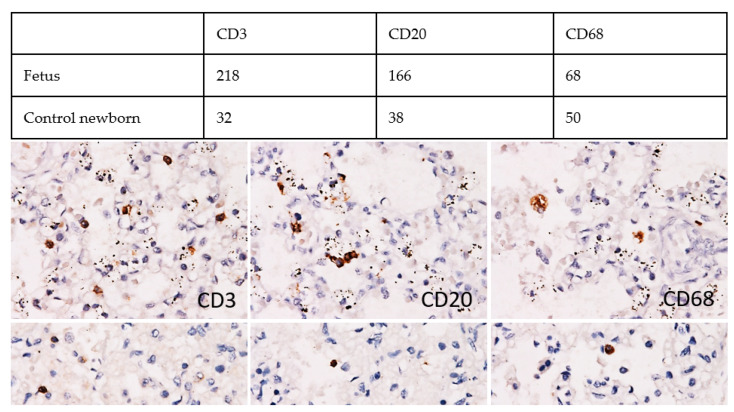
Inflammatory cell infiltration in lungs of the fetus infected with SARS-CoV-2 (top panel). There was a several-fold increase in T-cells (CD3) and B-cells (CD20), no difference in macrophages (CD68) infiltration when compared with control newborn lung tissue (lower panel). Immunoperoxidase, diaminobenzidine, 400×.

## Data Availability

Not Applicable.

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
