# Peer review of "Intrauterine Fetal Demise After Uncomplicated COVID-19: What Can We Learn from the Case?"

_viruses, 2021, doi:10.3390/v13122545_

Round 1

Reviewer 1 Report

The case described appears to be SARS-CoV2 associated fetal demise related to massive perivillous fibroinoid deposit /maternal floor infarction.  The association of massive perivillous fibrinoid deposit / maternal floor infarction with fetal demise and SARs-CoV2 infection has been described in the recent literature.  The pathologic diagnosis of massive perivillous fibrinoid deposit should be stated somewhere in the result section.  Also please cite the following recent relevant references and update the references:

  • DOI: 10.1093/jpids/piaa109
  •  10.1016/j.ajogmf.2020.100197
  • 10.1172/JCI139569
  • doi: 10.1177/10935266211020723.
  • https://doi.org/10.1038/s41379-021-00827-5
  • https://doi.org/10.1016/j.placenta.2021.07.288

Author Response

Thank you for your kind review.

- Perivillous fibrin deposition is stated in lin 143-150 and documented in Fig. 2.

- The requested works were cited in the text and included into the references

- Identified English grammar defects were corrected

With consideration,

P. Babál

Reviewer 2 Report

The authors report a case of apparent transplacental  transmission of SARS-CoV-2 associated with severe placental damage and IUFD at 38 weeks gestational age. Of note the infection in the mother was not acute and had only mild symptoms.

The manuscript is timely and has some merit, but many recent publications are not referenced and need to be included with some changes in the discussion. At least these three references need to be added and their recommendations for diagnosis and terminology followed: 

  • DOI: 10.3390/v12111308
  • DOI: 10.1016/j.ajog.2021.07.029
  • DOI.org/10.5858/arpa.2021-0246-SA

The authors need to state whether they detected replicating virus or not in order to make conclusions about active infection in the placenta.

It would be important, if possible, to determine which variant of  the virus was present.

Minor points: Was the placenta weight trimmed and fresh? Fixed? Untrimmed? 

English grammar errors, easy to fix with help from a native English speaker.

Author Response

Thank you for your kind review.

  • The statement about active viral infection was added in the lines 306, 326
  • The requested works were included into the references
  • The variant of the virus cannot be reported since at the time of the described case, no subtype specifications were performed
  • The placenta weight was that of the fixed organ before trimming.
  • English grammar errors were identified and corrected

With consideration,

P. Babal

Round 2

Reviewer 2 Report

The authors have addressed my concerns.